# CLDN6 Suppresses c–MYC–Mediated Aerobic Glycolysis to Inhibit Proliferation by TAZ in Breast Cancer

**DOI:** 10.3390/ijms23010129

**Published:** 2021-12-23

**Authors:** Huinan Qu, Da Qi, Xinqi Wang, Yuan Dong, Qiu Jin, Junyuan Wei, Chengshi Quan

**Affiliations:** The Key Laboratory of Pathobiology, Ministry of Education, College of Basic Medical Sciences, Jilin University, 126 Xinmin Avenue, Changchun 130021, China; quhn18@mails.jlu.edu.cn (H.Q.); qida19@mails.jlu.edu.cn (D.Q.); wangxq17@mails.jlu.edu.cn (X.W.); dongyuan20@mails.jlu.edu.cn (Y.D.); jinqiu21@mails.jlu.edu.cn (Q.J.); weijy20@mails.jlu.edu.cn (J.W.)

**Keywords:** CLDN6, breast cancer, proliferation, aerobic glycolysis, c–MYC

## Abstract

Claudin 6 (CLDN6) was found to be a breast cancer suppressor gene, which is lowly expressed in breast cancer and inhibits breast cancer cell proliferation upon overexpression. However, the mechanism by which CLDN6 inhibits breast cancer proliferation is unclear. Here, we investigated this issue and elucidated the molecular mechanisms by which CLDN6 inhibits breast cancer proliferation. First, we verified that CLDN6 was lowly expressed in breast cancer tissues and that patients with lower CLDN6 expression had a worse prognosis. Next, we confirmed that CLDN6 inhibited breast cancer proliferation through in vitro and in vivo experiments. As for the mechanism, we found that CLDN6 inhibited c–MYC–mediated aerobic glycolysis based on a metabolomic analysis of CLDN6 affecting cellular lactate levels. CLDN6 interacted with a transcriptional co–activator with PDZ-binding motif (TAZ) and reduced the level of TAZ, thereby suppressing c–MYC transcription, which led to a reduction in glucose uptake and lactate production. Considered together, our results suggested that CLDN6 suppressed c–MYC–mediated aerobic glycolysis to inhibit the proliferation of breast cancer by TAZ, which indicated that CLDN6 acted as a novel regulator of aerobic glycolysis and provided a theoretical basis for CLDN6 as a biomarker of progression in breast cancer.

## 1. Introduction

CLDNs are the most important tight junction (TJ) proteins that form paracellular barriers with gate and fence functions [1,2]. As a member of CLDNs, CLDN6 is involved in regulating epidermal barrier formation [3,4,5] and epithelial permeability [6]. In addition to the above canonical functions, dysregulated CLDN6 is related to neoplastic growth and aggressiveness in cancers [7]. We have reported that CLDN6 overexpression suppressed breast cancer metastasis by inhibiting EMT [8]. Additionally, CLDN6 can interact with proteins containing PDZ domains or PDZ–binding motifs (PBMs) such as ZO–1, AF–6, and β–catenin through its PBM, which made CLDN6 communicate with proteins or pathways to play important roles in breast cancers [8,9,10]. We have found that CLDN6 suppressed breast cancer cells proliferation [11], but the underlying mechanism is not well understood.

Sustaining proliferative signaling and reprogramming of energy metabolism are two hallmarks of cancer [12]. Aerobic glycolysis is the most important energy metabolism which allows cancer cells to meet energy and substances requirements for proliferation [13], and it is the main consequence of the aberrant activation of oncogenes like *c–MYC*, *mTOR*, and *HIF–1α* [14,15,16]. c–MYC promotes the aerobic glycolytic process by facilitating the transcription of target genes such as glucose transporter 1 (*GLUT1*) and lactate dehydrogenase A (*LDHA*) [17,18]. *c–MYC* mRNA was found down–regulated upon CLDN6 overexpression according to our previous RNAseq data of MCF–7 cells, but the validation and exploration of relevant mechanisms are still needed.

As a Hippo signaling nuclear effector, TAZ binds to the TEA–domain (TEAD) family of transcription factors in the nucleus, leading to increased cell proliferation [19]. *c–MYC* is a recognized target gene of the TAZ–TEAD transcription factor complex [20]. Simultaneously, TAZ is a PBM–containing protein that is crucial for the development of breast cancer [21,22]. Therefore, we speculated that CLDN6 may affect c–MYC by interacting with TAZ, and the mechanism by which CLDN6/TAZ/c–MYC mediated signaling controls glycolysis and proliferation in breast cancer has not been investigated to our knowledge.

Here, we reported that breast cancer patients with low expression of CDLN6 had a poor prognosis. CLDN6 inhibited the proliferation of breast cancer cells in vitro and in vivo. As for the mechanism, CLDN6 interacted with TAZ and reduced the TAZ level, which transcriptionally suppressed c–MYC expression, leading to a decrease of c–MYC–mediated aerobic glycolysis. In conclusion, our results supported that CLDN6 acted as a potential prognostic biomarker of breast cancer and provided new insights into the mechanism by which CLDN6 inhibited breast cancer proliferation.

## 2. Results

### 2.1. Low CLDN6 Expression Is Connected with a Poor Prognosis of Breast Cancer Patients

To verify the low expression of CLDN6 in breast cancer tissues, we analyzed the mRNA expression of CLDN6 with the GEO database GSE134359, which showed that the level of CLDN6 mRNA in breast tumor tissues was lower than that in normal tissues (Figure 1A). Next, we examined the CLDN6 protein expression in a TMA by IHC. Representative images of low and high CLDN6 staining were shown in Figure 1B. Almost two–thirds of tumor samples showed weak or absent CLDN6 expression, whereas positive CLDN6 staining was seen in about 50% of normal specimens (Figure 1C). We subsequently constructed the Kaplan–Meier curve of breast cancer patients using the GEO database (*n* = 2032, best cut–off), which showed that a lower CLDN6 level was associated with a worse relapse–free survival (RFS) (Figure 1D). This also applied when the patients were divided into basal–like, HER2+, luminal A, and luminal B subtypes (Figure 1E–H). Collectively, these data indicated that CLDN6 was reduced in breast cancer, and breast cancer patients with lower CLDN6 expression had a worse prognosis.

### 2.2. CLDN6 Suppresses Breast Cancer Cell Proliferation In Vitro and In Vivo

In light of former data, we further investigated the role of CLDN6 in breast cancer cells. Our group has established CLDN6–overexpressed MCF–7 and MDA–MB–231 cells, and we verified the high expression of CLDN6 by WB before the study (Appendix A). EdU assay results demonstrated that both MCF–7 and MDA–MB–231 cells with CLDN6 overexpression showed a lower proliferation rate versus controls (Figure 2A,B). Similarly, CLDN6 overexpression in both cells displayed lower cell viability (Figure 2C,D) and lower colony formation capacity than controls (Figure 2E,F). In order to further evaluate the effect of CLDN6 on in vivo tumorigenesis, we established xenograft implantation tumor models. As shown in Figure 2G–I, the tumors generated from MDA–MB–231/CLDN6 cells were smaller and lighter than those induced by MDA–MB–231/Vec cells. IHC staining of sections obtained from xenograft implantation tumors affirmed sustained CLDN6 expression in the membrane in the MDA–MB–231/CLDN6 cells–induced tumors along with inhibition of Ki67 expression versus the control group (Figure 2J,K). Taken together, the above results indicated that CLDN6 inhibited tumor growth in vitro and in vivo.

### 2.3. CLDN6 Induces Metabolomic Alterations in Breast Cancer Cells

To gain further mechanistic insight, we performed a metabolomics assay of MDA–MB–231 cells by the UHPLC–QTOFMS platform. As shown in Figure 3A–D, metabolic differences between the two groups (MDA–MB–231/Vec and MDA–MB–231/CLDN6 cells) in the positive and negative ion models were determined by a principal component analysis (PCA) and orthogonal partial least-squares discrimination analysis (OPLS–DA), suggesting that CLDN6 induced significant changes in the metabolite profile. We visualized the results of the screening for differential metabolites in the form of a volcano plot, which was shown in Figure 3E,F. Furthermore, a hierarchical cluster analysis showed that compared with the control group, 38 metabolites in the CLDN6 overexpression group were significantly changed in the positive ion mode, of which nine metabolites were increased and 29 reduced (Figure 3G). In the negative ion mode, there were 45 metabolites changed, with 22 metabolites up–regulated and 23 down-regulated (Figure 3H). Hence, our results implied that CLDN6 affected the metabolic process of MDA–MB–231 cells.

To better understand the specific role of CLDN6 in the metabolism, a pathway impact analysis was performed on individual metabolites using metabolic pathways from the Kyoto Encyclopedia of Genes and Genomes (KEGG). As shown in Figure 3I,J, the top altered metabolic pathways included purine metabolism, pyrimidine metabolism, glycine, serine and threonine metabolism, fatty acid biosynthesis metabolism, and pentose phosphate pathway.

### 2.4. CLDN6 Functions as a Negative Regulator of Aerobic Glycolysis in Breast Cancer Cells

In addition to the altered nucleotide, amino acid, and lipid metabolism, we also observed that lactate, a glycolytic by–product, was altered upon CLDN6 overexpression in Figure 3H. In terms of metabolic plasticity, we wanted to investigate whether CLDN6 regulated aerobic glycolysis in breast cancer cells. As shown in Figure 4A–C, CLDN6 overexpression decreased glucose uptake, lactate production, and ATP production in both MCF–7 and MDA–MB–231 cells. Moreover, we measured ECAR using the Seahorse extracellular flux analyzer to reflect aerobic glycolysis. As shown in Figure 4D–G, CLDN6 overexpression inhibited aerobic glycolysis in breast cancer cells. Hence, we assumed that CLDN6 suppressed aerobic glycolysis of breast cancer cells.

### 2.5. CLDN6 Down–Regulates c–MYC–Mediated Aerobic Glycolysis

According to our previous RNAseq data of MCF–7 cells, c–MYC mRNA expression was down–regulated after CLDN6 overexpression, so we assessed the effect of CLDN6 on c–MYC. As shown in Figure 5A–D, in MCF–7 and MDA–MB–231 cells, both the mRNA and protein level of c–MYC was notably decreased in the CLDN6 overexpression group compared with the control group. Similarly, c–MYC target genes GLUT1 and LDHA had similar alterations (Figure 5E–G). To further validate whether the suppression action of CLDN6 on glycolysis was dependent on c–MYC, we overexpressed c–MYC in MCF–7/CLDN6 and MDA–MB–231/CLDN6 cells and detected increased mRNA and protein levels of c–MYC, GLUT1, and LDHA (Figure 5H–J). Meanwhile, we observed that c–MYC overexpression reversed the suppression action of CLDN6 on glucose uptake and lactate production (Figure 5K,L), but caused a further decrease in ATP production (Figure 5M), which may be related to the altered cell proliferation induced by c–MYC. Overall, these results suggested that CLDN6 inhibited aerobic glycolysis via c–MYC.

### 2.6. CLDN6 Inhibits Breast Cancer Cell Proliferation via c–MYC In Vitro and In Vivo

In this regard, we evaluated the function of c–MYC in cell proliferation ability inhibited by CLDN6. EdU (Figure 6A,B), CCK8 (Figure 6C,D), and plate clone formation (Figure 6E,F) results confirmed that c–MYC reversed the suppression action of CLDN6 on cell proliferation. In line with the observations in vitro, in vivo experiments showed that both volume and weight of implantation tumors induced by MDA–MB–231/CLDN6 with c–MYC overexpression cells have increased significantly versus the control group (Figure 6G–I). In addition, c–MYC overexpression profoundly increased cell proliferation in tumors as demonstrated by the staining of Ki67 (Figure 6J,K). In summary, these results suggested that the inhibitory effect of CLDN6 on breast cancer growth was c–MYC–dependent.

### 2.7. CLDN6 Inhibits the Transcription of c–MYC by Interacting with TAZ

In terms of the mechanism by which CLDN6 affected c–MYC, we focused on YAP and TAZ, which are not only the effector molecules containing PBMs in the Hippo pathway but also the transcriptional co–activators of c–MYC [20]. We noted the effect of CLDN6 on the protein expression of YAP and TAZ in MCF–7 and MDA–MB–231 cells. A WB analysis showed that CLDN6 significantly inhibited TAZ protein expression (Figure 7A,B), but had no significant influence on YAP protein expression in both MCF–7 and MDA–MB–231 cells (Appendix A). We next observed the cellular localization of TAZ in MCF–7 and MDA–MB–231 cells with CLDN6 overexpression by the IF assay. As shown in Figure 7C, more cytoplasmic and less nuclear localization of TAZ expression was seen upon CLDN6 overexpression in both MCF–7 and MDA–MB–231 cells. Importantly, we noticed a marked co–localization of TAZ and CLDN6 in the membrane of MDA–MB–231 cells. Our Co–IP results validated the association of CLDN6 with TAZ (Figure 7D). Taken together, we demonstrated that CLDN6 kept TAZ in the cytoplasm or membrane by interacting with TAZ, leading to a reduction in nuclear TAZ, which may be one of the mechanisms by which CLDN6 inhibited c–MYC transcription.

### 2.8. The Expression of CLDN6, TAZ, and c–MYC Expression in Breast Cancer Patients and Xenografts Tissues in Nude Mice

To evaluate the relationship between CLDN6/TAZ/c–MYC expression and clinicopathological parameters, we did a TMA–IHC analysis of breast cancer patients (Figure 8A). As shown in Table 1, patients with high expression of c–MYC were likely to develop lymph node metastasis. In our study, there was no association between CLDN6 or TAZ expression and clinical characteristics, which seems to be explained by the limited sample size. There is no significant correlation between CLDN6 and TAZ or c–MYC in breast cancer tissues shown by a Pearson correlation analysis (Figure 8B,C), whereas c–MYC was significantly positively correlated with TAZ (Figure 8D). In addition, representative IHC images of CLDN6, TAZ, c–MYC, GLUT1, and LDHA in the transplanted tumor tissues of nude mice (resulting from MDA–MB–231/CLDN6 and control cells) were shown in Figure 8E.

## 3. Discussion

More than a simple static component that establishes the tight junction, CLDN6 is also a cellular signaling component that functions to receive environmental signals and transmit them intracellularly. It is relevant to multiple biological behaviors of tumors such as proliferation, apoptosis, autophagy, drug resistance, and metastasis [7,23]. CLDN6 is an ideal target for antibody approaches of high potency in cancers with high CLDN6 expression based on the fact that CLDN6 is undetectable in the adult tissues [24,25,26]. Notably, CLDN6 exerts carcinogenic or anti–carcinogenic effects depending on the tissue or cell type [27]. As for proliferation, CLDN6 promotes cell proliferation in human hepatocellular carcinoma, gastric cancer, and endometrial carcinoma [28,29,30,31]. However, we have found the proliferation of MCF–7 breast cancer cells was suppressed upon CLDN6 overexpression, but the mechanism was unknown. In this study, we focused on the expression and prognosis of CLDN6 in breast cancer patients and explored the mechanism by which CLDN6 inhibited breast cancer proliferation.

The expression of CLDN6 was lower in breast cancer tissues than non–neoplastic tissues based on a GEO dataset analysis and TMA analysis, and its low expression was associated with a poor prognosis, which is in line with the study by Jia et al. [32]. Our previous study reported that a low level of CLDN6 in breast cancer was due to DNA methylation, but the reason for low CLDN6 expression in breast cancer warrants further study [33].

In vitro and in vivo experiments showed that CLDN6 overexpression inhibited breast cancer proliferation. The reprogramming of energy metabolism is a hallmark of cancer, which has an important impact on tumor proliferation [34]. Currently, there is very limited research on CLDNs and cellular metabolism. It is reported that lactate dehydrogenase B (LDHB) suppression leads to hepatoma cell invasiveness via inducing CLDN1 expression [35]. When studying the relationship between mitochondrial ribosomal protein L13 (MRPL13) and CLDN1 in liver cancer patients, Lee YK et al. [36] found that in the MRPL13–low group, CLDN1 expression negatively correlated with pyruvate dehydrogenase B (PDHB) expression and the ratio of LDHB/LDHA, which suggested a metabolic change to glycolysis. Our metabolomic analysis showed that CLDN6 significantly altered the metabolism of MDA–MB–231 cells. Given the apparent alteration of lactate and the plasticity aspect of metabolism, we assumed that CLDN6 might affect the aerobic glycolysis process. Glycolysis, glycolytic capacity, and glycolytic reserve were found decreased upon CLDN6 overexpression in breast cancer cells by measuring ECAR. Of course, altered metabolic pathways upon CLDN6 overexpression such as purine, pyrimidine, and fatty acid biosynthesis metabolism may also contribute to the inhibitory effect of CLDN6 on tumor growth, which is our ongoing research.

Tumor metabolic dysregulation is usually achieved by gene amplification or deletion or the altered activation status of upstream growth signaling pathways [37]. c–MYC is an oncogenic transcription factor that is commonly overexpressed and is one of the key proteins for aerobic glycolysis in many human tumors [14]. We found that CLDN6 overexpression inhibited mRNA and protein expression of c–MYC and target glycolytic genes *GLUT1* and *LDHA*. Similarly, Ji H et al. [38] found that CLDN7 knockdown increased the c–MYC protein expression in salivary adenoid cystic carcinoma, and Che J et al. [39] found that overexpression of CLDN3 reduced c–MYC expression.

*c–MYC* is one of the target genes of the TEAD, a key transcription factor of YAP/TAZ. There are several studies about CLDNs and YAP/TAZ. CLDN18 interacts with p–YAP to inhibit YAP nuclear localization in progenitors of distal lung epithelium AT2 cells [40]. Dependent on the PBM, CLDN2 binds to and regulates the nuclear expression and activation of YAP [41]. The nuclear expression and activity of TAZ are also regulated by the upstream kinases of the Hippo pathway and other kinases such as c–SRC [42,43]. In gastric cancer cells, CLDN6 was reported to interact with LATS1/2 and reduce the level of p–YAP, thereby increasing the nuclear YAP activity [28]. In our study, CLDN6 interacted with TAZ, inhibiting its nuclear expression, but the details of their interaction and the effect on TAZ degradation need to be further investigated. In addition, we cannot eliminate the potential effects of CLDN6 on other proteins in the Hippo pathway, which is one of our future studies.

In summary, we found that CLDN6 was low-expressed in breast cancer, and CLDN6 overexpression inhibited cell proliferation in vitro and in vivo. Mechanistically, CLDN6 retained TAZ in the cytoplasm by interacting with it, which led to a reduction of TAZ, especially in the nucleus. Then, the transcription of c–MYC and the expression of its target genes GLUT1 and LDHA were inhibited, ultimately leading to decreased aerobic glycolysis and cell proliferation. Our findings contribute to the understanding of the important role of CLDN6 in regulating breast cancer metabolism, especially aerobic glycolysis. CLDN6 has the possibility to be used as a biomarker for the diagnosis and prognosis of breast cancer.

## 4. Materials and Methods

### 4.1. Immunohistochemistry (IHC)

Human breast cancer tissue microarray (TMA; HBreD070CS02) was purchased from Shanghai Outdo Biotech CO. Paraffin sections were dehydrated in xylene, anhydrous ethanol, 95% ethanol, and 85% ethanol and then subjected to antigen repair, followed by treatment with the UltraSensitive™ SP (Mouse/Rabbit) IHC Kit (KIT–9710, MXB Biotechnology, Fuzhou, China). The first antibodies are listed in Appendix A.

IHC scores are a combination of positive intensity (0, none; 1, weak; 2, moderate; 3, strong) and a proportion of positive cells (≤25%: 1; 26–50%: 2; 51–75%: 3; ≥75%: 4). IHC staining was classified as either high expression (score ≥ 8) or low expression (score < 8).

### 4.2. Cell Culture

MCF–7, MDA–MB–231, and HEK293T were cultured in Dulbecco’s modified Eagle’s medium (DMEM; Meilune, Dalian, China) containing 10% fetal bovine serum (FBS; Gibco, Carlsbad, CA, USA) at 37 °C in a humidified incubator containing 5% CO_2_.

### 4.3. Transfection

The CLDN6–GFP–luciferase overexpression lentivirus was constructed by Genechem (Shanghai, China). The c–MYC overexpression plasmid was generated by PPL Genebio Technology (Nanjing, China). The transfection process was performed as described by Yang et al. [44].

### 4.4. RNA Extraction and RT–PCR

Total RNA was isolated using the TRIzol reagent (Invitrogen, Carlsbad, CA, USA), and was reverse transcribed into cDNA using MonaScript RT All–in–One Mix with dsDNase (Monad, Wuhan, China). The PCR products were electrophoresed and subjected to autoradiography. Primers were synthesized by Sangon (Shanghai, China) and listed in Appendix A.

### 4.5. Western Blot (WB)

A WB assay was performed as previously described [8]. The first antibodies used in this study are listed in Appendix A.

### 4.6. EdU Assay

EdU assay was performed using the EdU–555 Cell Proliferation Assay Kit (Beyotime, Shanghai, China). Observation and photography were performed using a fluorescence microscope (Olympus, Tokyo, Japan).

### 4.7. Cell Counting Kit–8 (CCK8)

Cell viability was measured using CCK8 reagents (Meilune). The cells were seeded in 96–well plates at the density of 1000 cells/well and incubated for 24 h. A 1:9 diluted CCK8 solution in DMEM was added into each well at 1 d, 2 d, 3 d, 4 d, and 5 d, and the plates were incubated at 37 °C for 2 h in 5% CO_2_. The absorbance of the sample was measured at 450 nm on a microplate reader (Thermo, Schwerte, Germany).

### 4.8. Plate Clone Formation Assay

The cells were plated into six-well plates at 600 cells/well in triplicate. The medium was replaced every 2–3 days with a fresh medium until colonies were visible with naked eyes. The colonies were washed and fixed, stained with 5% crystal violet, and then counted.

### 4.9. In Vivo Tumor Xenograft Model

A total of 20 female BALB/cA–nu mice (4 weeks old, 16–20 g, specific pathogen–free standard) were purchased from Beijing Huafukang Company and maintained in the Animal Experiment Center of the School of Basic Medicine, Jilin University (Laboratory Animal Use License No.: SYXK (Jilin) 2018-0001). The protocol and procedures employed were ethically reviewed and approved by the Experimental Animal Ethical Committee of Jilin University. All animal experiments were performed in accordance with relevant institutional and national guidelines and regulations for the care and use of laboratory animals. Each nude mouse was inoculated with 5 × 10^6^ cells in 100 µL PBS subcutaneously. The length (L) and width (W) of the tumor volume (V) were measured every four or five days. The tumor volume was estimated by the formula V = 0.5 × L × W^2^. The transplanted tumors were fixed in 10% neutral formalin and stained for IHC.

### 4.10. Sample Collection and Metabolomics Assay

We collected and quenched the cells in liquid nitrogen for 4 min, and stored at −80 °C. Then the cells were transported on dry ice to Biotree Biotech Co., Ltd. (Shanghai, China) to perform a metabolomics assay.

### 4.11. Aerobic Glycolysis Measurement

A Glucose Assay Kit and Lactate Assay Kit (Jiancheng Bioengineering Institute, Nanjing, China), and ATP Assay Kit (Beyotime) were used to test the glucose uptake, lactate production, and ATP production separately. Glycolysis was also measured by examining the extracellular acidification rate (ECAR) using a Seahorse extracellular flux analyzer in Baihaobio Co., Ltd. (Benxi, China).

### 4.12. Immunofluorescence (IF)

Cells were fixed with 4% paraformaldehyde and then incubated with 0.5% Triton X–100 and 5% BSA. After being incubated with the first antibody at 4 °C overnight and following the secondary antibody, the cells were stained with DAPI and visualized with a fluorescence microscope (Olympus). The first antibodies used in IF were listed in Appendix A.

### 4.13. Co–Immunoprecipitation (Co–IP) Assay

The Co–IP assay was performed as previously described [45]. An anti–CLDN6 (E2S5M, Cell Signaling Technology, Danvers, MA, USA) antibody was used for IP.

### 4.14. Statistical Analysis

All statistical analyses were performed using GraphPad Prism 8.0 (GraphPad, San Diego, CA, USA). The data were presented as the mean ± standard deviation (SD) of at least three independent experiments. Data were analyzed using Student’s *t*-test for comparison between groups. The protein expression levels and clinical characteristics were compared by the chi-square test. Correlations between proteins were analyzed using Pearson’s correlation coefficients. *p* < 0.05 was considered statistically significant.

## Figures and Tables

**Figure 1 ijms-23-00129-f001:**
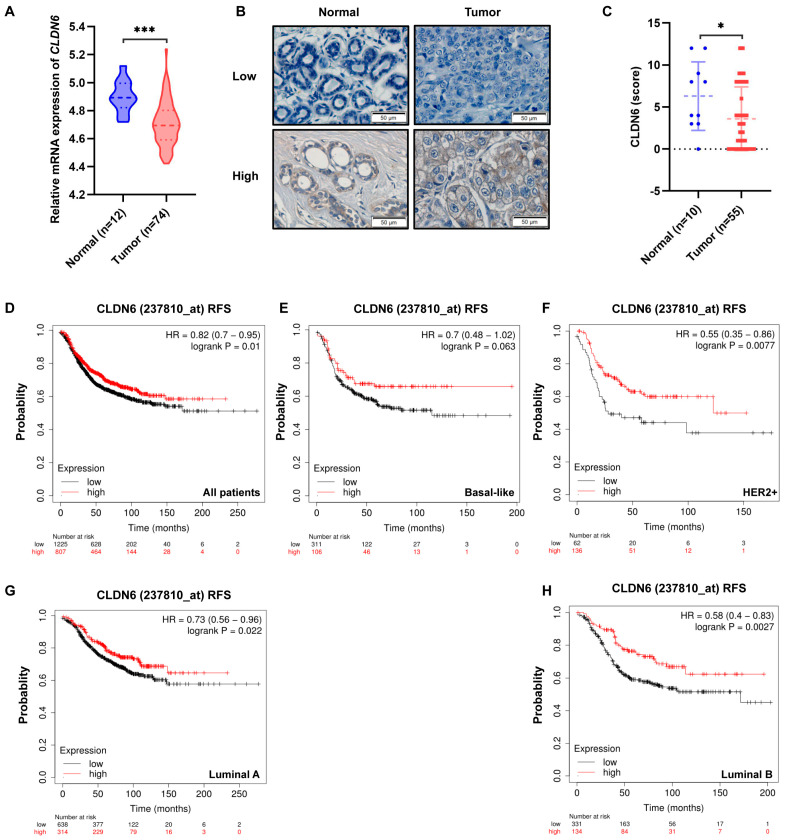
Expression of CLDN6 in breast cancer tissues and its relationship with patient prognosis. (**A**) The mRNA expression of CLDN6 in adjacent normal and breast cancer tissues in the GEO database GSE134359. *** *p* < 0.001. (**B**) Representative graphs of CLDN6 IHC staining for high and low expression in TMA. Scale bar, 50 µm. (**C**) Analysis of the staining intensity score of CLDN6 in TMA. * *p* < 0.05. (**D**) The relationship between CLDN6 expression and RFS in 2032 breast cancer patients analyzed by Kaplan–Meier plotter. Kaplan–Meier survival curves of CLDN6 expression and RFS in basal–like (**E**), HER2+ (**F**), luminal A (**G**), and luminal B (**H**) breast cancer patients.

**Figure 2 ijms-23-00129-f002:**
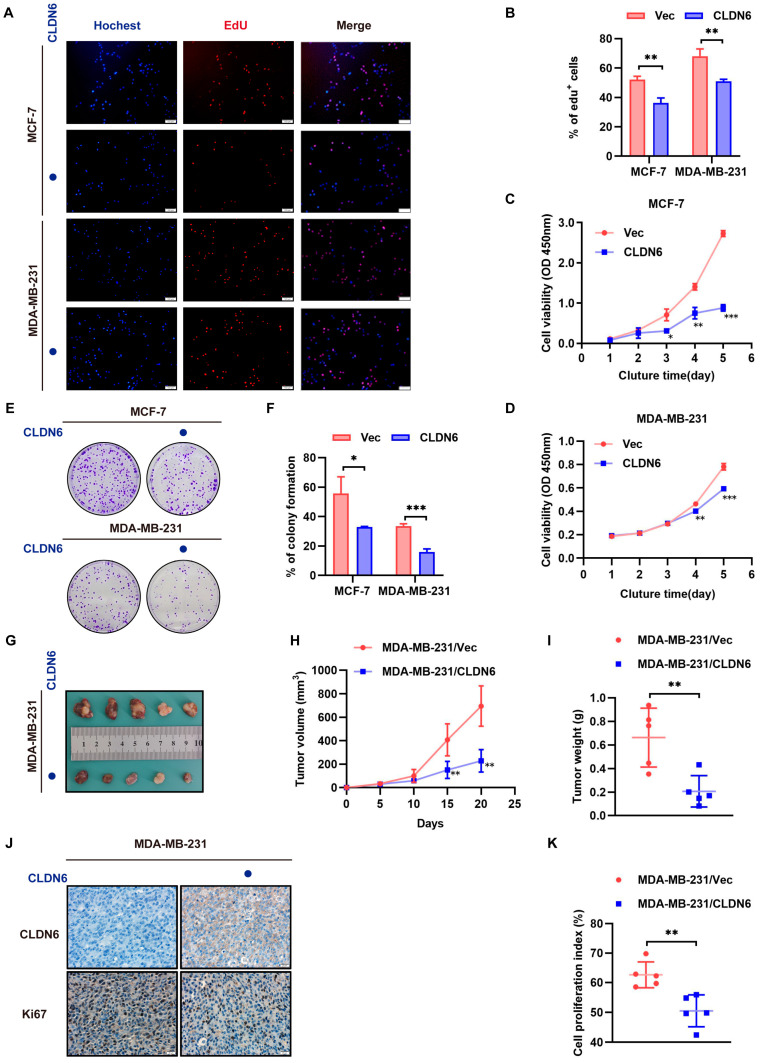
CLDN6 inhibited breast cancer cells proliferation in vitro and in vivo. (**A**,**B**) Cell proliferation ability was determined by EdU. Scale bar, 100 µm. (**C**,**D**) CCK8 assay to assess cell viability. (**E**,**F**) Clone formation assay. (**G**) Images of subcutaneous xenograft tumors (5 per group). Tumor volume (**H**) and weight (**I**) was measured. (**J**) Representative images of CLDN6 and Ki67 IHC staining. Scale bar, 20 µm. (**K**) Cell proliferation index between MDA–MB–231/CLDN6 and the control groups * *p* < 0.05, ** *p* < 0.01, *** *p* < 0.001. 
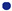
 CLDN6 overexpression.

**Figure 3 ijms-23-00129-f003:**
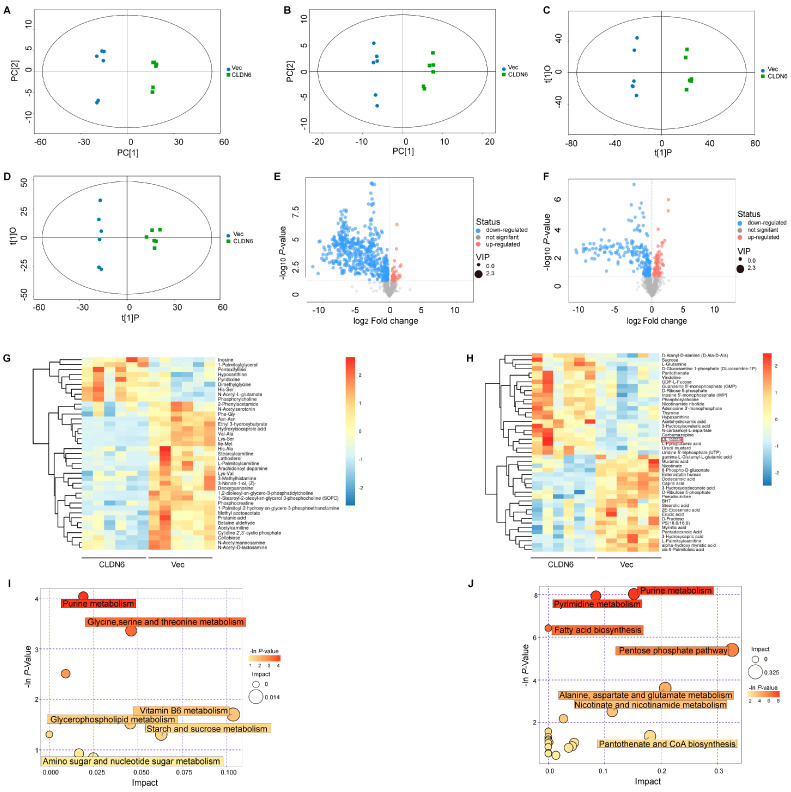
CLDN6 regulates the metabolism of MDA−MB−231 cells. PCA and OPLS–DA plots generated from LC−MS/MS data of cellular metabolites in the positive (**A**,**C**) and negative (**B**,**D**) ion mode. (**E**,**F**) The volcano plot of differential metabolites between MDA−MB−231/Vec and MDA−MB−231/CLDN6 cells in the positive (left) and negative (right) ion mode. (**G**,**H**) Heatmap of hierarchical clustering analysis for group MDA−MB−231/CLDN6 vs. MDA−MB−231/Vec in the positive (left) and negative (right) ion mode. (**I**,**J**) Bubble chart of metabolic pathway analysis in the positive (left) and negative (right) ion mode. The color of the circle represents the *p* value, and the size represents the pathway impact value.

**Figure 4 ijms-23-00129-f004:**
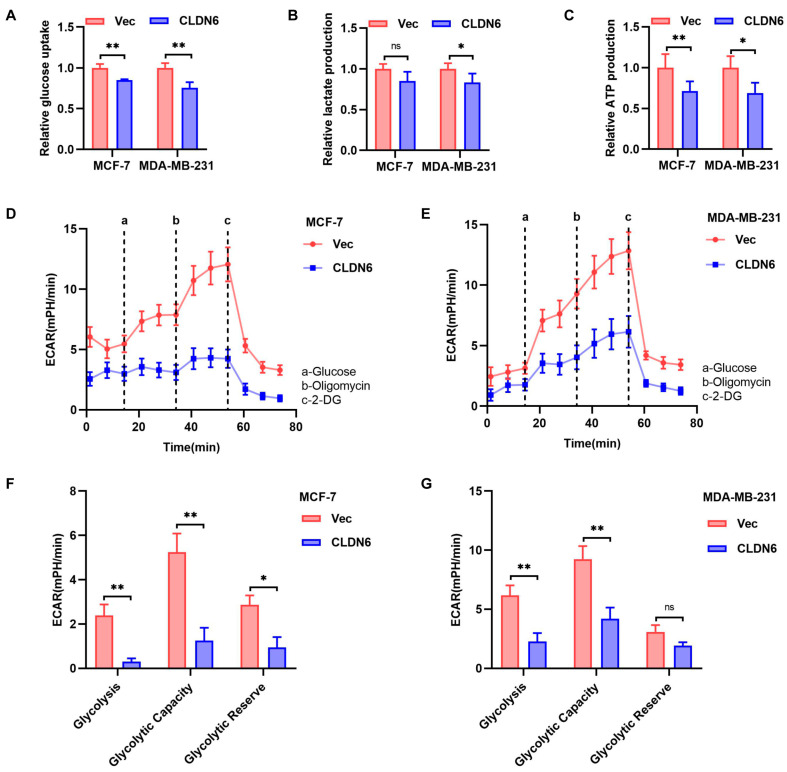
CLDN6 inhibits aerobic glycolysis in MCF–7 and MDA–MB–231 cells. Glucose uptake assays (**A**), lactate production assays (**B**), and ATP production assays (**C**) with MCF–7 and MDA–MB–231 cells expressing vector or CLDN6. (**D**,**E**) Diagram of ECAR results of MCF–7/CLDN6, MDA–MB–231/CLDN6, and their controls were obtained by Seahorse extracellular flux analyzer. (**F**,**G**) Comparison of glycolysis, glycolytic capacity, and glycolytic reserve in MCF–7/CLDN6 and MDA–MB–231/CLDN6 and the respective control cells. Glycolysis = (maximum rate measurement before Oligomycin injection) − (last rate measurement before glucose injection); Glycolytic capacity = (maximum rate measurement after Oligomycin injection) − (last rate measurement before glucose injection); Glycolytic reserve = (Glycolytic capacity) − (Glycolysis). * *p* < 0.05, ** *p* < 0.01.

**Figure 5 ijms-23-00129-f005:**
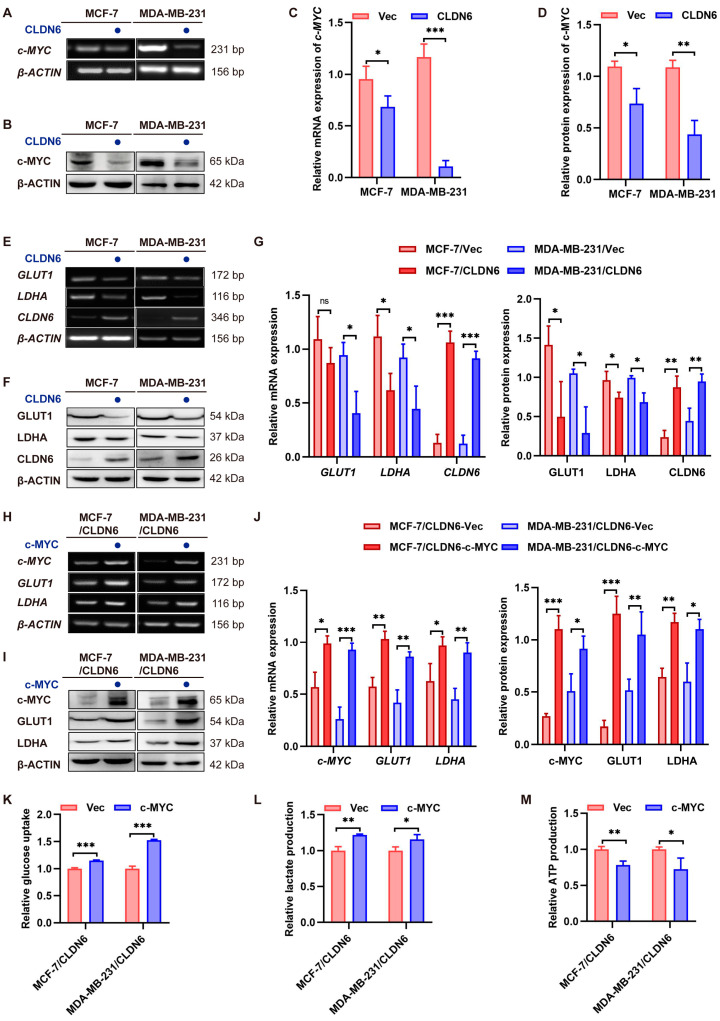
CLDN6 inhibits aerobic glycolysis by down-regulating c–MYC and its target genes GLUT1 and LDHA. RT-PCR (**A**,**B**) and WB (**C**,**D**) analysis of the impact of CLDN6 on c–MYC in MCF–7 and MDA–MB–231 cells. (**E**–**G**) RT–PCR and WB analysis of the effect of CLDN6 on GLUT1 and LDHA in MCF–7 and MDA–MB–231 cells. (**H**–**J**) RT–PCR and WB analysis of the effect of c–MYC on GLUT1 and LDHA in MCF–7/CLDN6 and MDA–MB–231/CLDN6 cells. Glucose uptake assays (**K**), lactate production assays (**L**), and ATP production assays (**M**) with MCF–7/CLDN6 and MDA–MB–231/CLDN6 cells expressing vector or c–MYC. ns, no significance, * *p* < 0.05, ** *p* < 0.01, *** *p* < 0.001. 
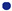
 CLDN6 or c–MYC overexpression.

**Figure 6 ijms-23-00129-f006:**
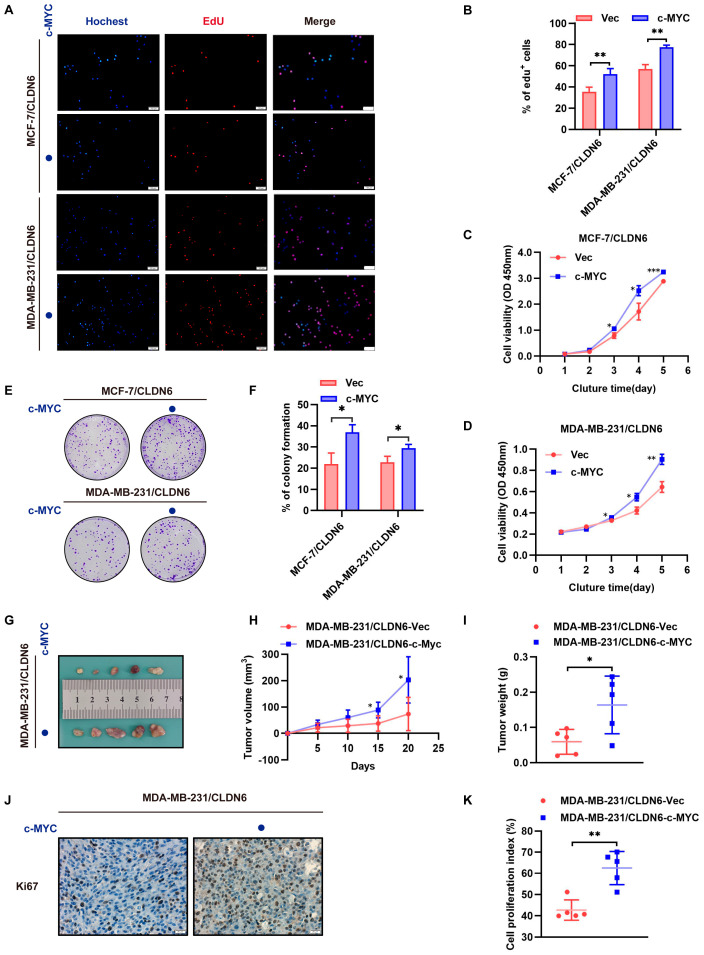
CLDN6 inhibits breast cancer cell proliferation via c–MYC in vitro and in vivo. (**A**,**B**) Cell proliferation ability was determined by EdU. Scale bar, 100 µm. CCK8 assay (**C**,**D**) to assess the effect of c–MYC on cell viability and clone formation assay (**E**,**F**) to assess the effect of c–MYC on cell proliferation in MCF–7/CLDN6 and MDA–MB–231/CLDN6 cells. (**G**) Subcutaneous xenograft tumor growth in nude mice (5 per group) was measured and compared in tumor volume (**H**) and weight (**I**). (**J**) Representative images of IHC staining of Ki67 of tumor sections. Scale bar, 20 µm. (**K**) Cell proliferation index was determined by the proportion of nuclear Ki67-positive cells. * *p* < 0.05, ** *p* < 0.01, *** *p* < 0.001. 
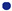
 c–MYC overexpression.

**Figure 7 ijms-23-00129-f007:**
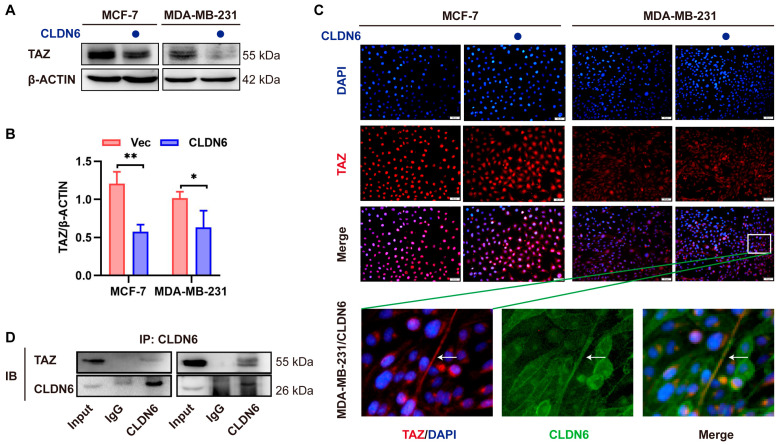
CLDN6 inhibits the transcription of c–MYC via TAZ. (**A**,**B**) WB analysis of CLDN6 on the protein expression of TAZ in MCF–7 and MDA–MB–231 cells. * *p* < 0.05, ** *p* < 0.01. (**C**) IF analysis of the effect of CLDN6 on TAZ in MCF–7 and MDA–MB–231 cells. Scale bar, 50 µm. (**D**) The interaction of CLDN6 and TAZ was detected by Co–IP assay in MCF–7/CLDN6 and MDA–MB–231/CLDN6 cells. 
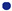
 CLDN6 overexpression.

**Figure 8 ijms-23-00129-f008:**
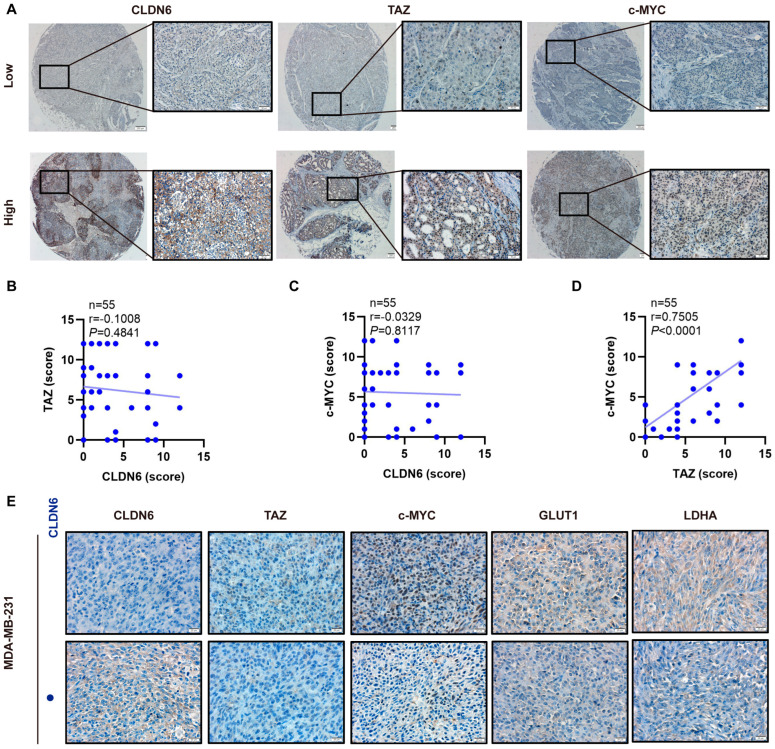
Expression of CLDN6, TAZ, and c–MYC in human breast cancer tissues and subcutaneous xenograft tumor tissues from nude mice. (**A**) Representative IHC images of low and high CLDN6, TAZ, and c–MYC expression in breast cancer tissues. Scale bar, 200 µm (left), 50 µm (right). Correlation analysis between CLDN6 and TAZ (**B**), CLDN6 and c–MYC (**C**), and TAZ and c–MYC (**D**) in breast cancer tissues. (**E**) Representative IHC images of CLDN6, TAZ, c–MYC, GLUT1, and LDHA expression in subcutaneous xenograft tumor tissues resulting from MDA–MB–231/Vec and MDA–MB–231/CLDN6 cells. Scale bar, 20 µm. 
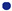
 CLDN6 overexpression.

**Table 1 ijms-23-00129-t001:** The association between CLDN6, TAZ, and c–MYC expression and clinical characteristics of breast cancer patients.

Clinicopathologic Features	CLDN6			TAZ			c–MYC		
	Low	High	*p*	Low	High	*p*	Low	High	*p*
Age			0.2048			0.1428			0.1517
≤57 years	21	5		11	15		13	13	
>57 years	19	10		18	11		20	9	
Tumor size			0.6504			0.5509			0.4277
≤3 cm	24	10		19	15		19	15	
>3 cm	16	5		10	11		14	7	
Lymph node			0.2164			0.3777			0.0364 *
Negative	22	11		19	14		20	7	
Positive	18	4		10	12		13	15	
Pathology grade			0.8257			0.8984			0.3217
I–II	20	7		14	13		18	9	
II–III	20	8		15	13		15	13	
Tumor stage			0.1523			0.8984			0.6596
0–IIA	18	10		15	13		16	12	
IIB–III	22	5		14	13		17	10	

* *p* < 0.05.

## Data Availability

The data presented in this study are available on request from the corresponding author on reasonable request.

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
