# Peer review of "CLDN6 Suppresses c–MYC–Mediated Aerobic Glycolysis to Inhibit Proliferation by TAZ in Breast Cancer"

_ijms, 2021, doi:10.3390/ijms23010129_

Round 1
Reviewer 1 Report
In the present manuscript, Qu et al showed that CLDN6 overexpression inhibited c-MYC mediated aerobic glycolysis to cancer cell proliferation by TAZ and acts as a biomarker of breast cancer progression. The manuscript is original research conducted with logical experiments and the findings are well supported by the existing literature, there are a few concerns that are listed below.
In Fig 7, the author showed that CLDN6 overexpression significantly inhibited TAZ protein expression, but there is no change in TAZ expression after overexpression Fig 7C. It would be nice to include the MCF7 inset in the images.
-
It would be nice if authors include apoptosis/autophagy marker staining in the tumor xenograft section.
-
The author should explain why they used MD-MB231 for invivo studies.
Reviewer 2 Report
In this article, Qu et al. give mechanistic insights into CLDN6 mediated inhibition of breast cancer progression. The study suggests that CLDN6 suppresses c-myc mediated aerobic glycolysis inhibiting breast cancer proliferation mediated by TAZ. The study is sound and interesting and substantiates development of CLDN6 as a breast cancer progression marker.
I have few comments to make:
- Overall the paper is well written but I believe the abstract can be polished further for English language. For e.g. the very first sentence of the abstract looks incoherent. It should be “Claudin-6 (CLDN6) was found to be………”.
- It is nowhere written in the manuscript whether the breast cancer patients were on any therapy. My concern is if the patients were given drugs or any kind of therapeutic treatment, it could have effects on the outcome of the study.
- Authors should also discuss if CLDN6 can also be applied as a biomarker for triple-negative breast cancer (TNBC) progression.
Reviewer 3 Report
The study by Wang et al. presents the role of CLDN6 in the proliferation of breast cancer cells. The study is nicely planned, the execution is extremely well performed and the presentation of the manuscript is done perfectly. The only thing which I believe needs to be improved is discussion. Now, it is limited to several statements and deserves to be supplemented with a nice graphical presentation of conclusions.
Round 2
Reviewer 3 Report
The Authors improved theirmanuscript